# Analysis of Microorganism Colonization, Biofilm Production, and Antibacterial Susceptibility in Recurrent Tonsillitis and Peritonsillar Abscess Patients

**DOI:** 10.3390/ijms231810273

**Published:** 2022-09-07

**Authors:** Renata Klagisa, Karlis Racenis, Renars Broks, Arta Olga Balode, Ligija Kise, Juta Kroica

**Affiliations:** 1Department of Otorhinolaryngology, Daugavpils Regional Hospital, LV-5401 Daugavpils, Latvia; 2Department of Doctoral Studies, Riga Stradins University, LV-1007 Riga, Latvia; 3Department of Biology and Microbiology, Riga Stradins University, LV-1007 Riga, Latvia; 4Center of Nephrology, Pauls Stradins Clinical University Hospital, LV-1002 Riga, Latvia; 5Department of Microbiology, NMS Laboratory, LV-1039 Riga, Latvia

**Keywords:** biofilm, recurrent tonsillitis, peritonsillar abscess

## Abstract

Background: Despite the widespread use of antibiotics to treat infected tonsils, episodes of tonsillitis tend to recur and turn into recurrent tonsillitis (RT) or are complicated by peritonsillar abscesses (PTAs). The treatment of RT and PTAs remains surgical, and tonsillectomies are still relevant. Materials and methods: In a prospective, controlled study, we analyzed the bacteria of the tonsillar crypts of 99 patients with RT and 29 patients with a PTA. We performed the biofilm formation and antibacterial susceptibility testing of strains isolated from study patients. We compared the results obtained between patient groups with the aim to identify any differences that may contribute to ongoing symptoms of RT or that may play a role in developing PTAs. Results: The greatest diversity of microorganisms was found in patients with RT. Gram-positive bacteria were predominant in both groups. *Candida* species were predominant in patients with a PTA (48.3% of cases). Irrespective of patient group, the most commonly isolated pathogenic bacterium was *S. aureus* (in 33.3% of RT cases and in 24.14% of PTA cases). The most prevalent Gram-negative bacterium was *K. pneumoniae* (in 10.1% of RT cases and in 13.4% of PTA cases). At least one biofilm-producing strain was found in 37.4% of RT cases and in 27.6% of PTA cases. Moderate or strong biofilm producers were detected in 16 out of 37 cases of RT and in 2 out of 8 PTA cases. There was a statistically significant association found between the presence of Gram-positive bacteria and a biofilm-formation phenotype in the RT group and PTA group (Pearson χ^2^ test, *p* < 0.001). *S. aureus* and *K. pneumoniae* strains were sensitive to commonly used antibiotics. One *S. aureus* isolate was identified as MRSA. Conclusions: *S. aureus* is the most common pathogen isolated from patients with RT, and *Candida* spp. are the most common pathogens isolated from patients with a PTA. *S. aureus* isolates are susceptible to most antibiotics. Patients with RT more commonly have biofilm-producing strains, but patients with a PTA more commonly have biofilm non-producer strains. *K. pneumoniae* does not play a major role in biofilm production.

## 1. Introduction

Recurrent tonsillitis (RT) is the repetitive inflammation of the palatine tonsils predominantly, or even exclusively, caused by bacteria [1,2]. Episodes of tonsillitis are characterized by fever, sore throat, odynophagia, congested tonsils with or without exudate, and cervical lymphadenopathy [1]. RT can be diagnosed clinically on an anamnestic report [3]. It can be considered when more than two distinct episodes of tonsillitis are encountered within a 12-month period [4]. Episodes of tonsillitis are treated with antibiotics [3,5]. Tonsillectomy is recommended for patients with RT who have experienced at least seven attack episodes per year in the preceding one year, five episodes per year in the preceding two years, or three episodes per year in the preceding three years despite adequate antibiotic therapy [6].

Inflammation from the palatine tonsil can be transmitted to adjacent peritonsillar tissue and form an abscess in peritonsillar space [3]. A peritonsillar abscess (PTA) is the most common purulent complication of acute tonsillitis. The diagnosis of PTA is based on medical history combined with a general clinical assessment. A patient with a PTA typically presents with fever, a sore throat, unclear speech, sometimes trismus, or a reaction of the descending lymph nodes. Clinically, redness and arching of the palpably stiff and markedly painful soft palate are found [7]. A peritonsillar abscess is typically unilateral. A PTA requires antibiotics and surgical management—incision and drainage or immediate tonsillectomy must be performed [8].

The surgical removal of tonsils or tonsillectomy remains a common operation. For example, in Germany, in 2013, a total of 84,332 patients underwent extracapsular tonsillectomies, and approximately 12,000 surgical procedures in terms of abscess tonsillectomies or incision and drainage were performed for patients with a PTA [8]. Episodes of tonsillitis decrease the quality of life and are a financial burden due to school or work absences and health care costs [8]. The disadvantages of antibiotic therapy are the promotion of bacterial resistance and the surgical procedures that expose patients to surgical and anesthetic risks [8].

The explanations of unsuccessful antibiotic therapy are, for example, difficulty in identifying causative bacteria, low concentrations of the antibiotics in the tonsillar tissue, or the specific antibiotic resistance patterns of the involved pathogenic bacteria or biofilm formation [9,10]. Several studies have been conducted to elucidate the spectra of pathogens involved in RT and PTAs [2]. Several pathogens with varying proportions are implicated in tonsil infections, including group A beta hemolytic *Streptococcus*; alpha hemolytic *Streptococcus*, *Hemophilus influenzae*; *Staphylococcus aureus*; *Enterococcus* spp.; *Klebsiella pneumoniae*; *Moraxella catarrhalis*; *Corynebacterium* spp.; and anaerobes, such as *Peptostreptococci*, *Fusobacterium*, *Veillonella*, and *Prevotella* [3]. A microbiological analysis of tonsils is challenging due to the difficulties in distinguishing between commensal and pathogenic germs, great diversity, and differences in the normal microbiota between patients [3,11]. In previous studies, the tonsils were found to have the greatest microbial diversity, which varied significantly among subjects [12,13]. As cultures are obtained from an area that is normally heavily colonized, the etiological relevance of each bacterium is raised [14]. The tonsillar surface is colonized by a normal oral microbiota, which is not usually implicated in tonsil infections [15]. However, autoinfection via the normal flora of the mouth and the pharynx is also possible [3]. Tonsillar infection may stem from bacteria within tonsillar crypts or the parenchyma rather than from those on the surface [16]. Crypts are narrow passages that penetrate the tonsils. Therefore, samples from tonsillar crypts are considered more appropriate for microbiological testing than tonsillar surface swabbing [11,15,17].

Studies have been performed comparing the results of microbiological analyses between patients with recurrent tonsillitis and healthy subjects, but little differences were found between the study participants [18,19,20,21]. There are many studies that analyze the microbiological results of PTAs without comparisons with other patient groups [7,22,23,24,25]. Not so many studies concerning RT and PTA are available, and the reason for the lack of success of conservative therapeutic approaches is not well understood [2].

Studies show a high prevalence of *S. aureus* in tonsillar samples from patients with recurrent tonsillitis [2,26]. *S. aureus* is considered the main etiological factor of RT. However, the role of this pathogen in the pathogenesis of RT exacerbation, in the formation of abscesses, and in the resistance to antibacterial therapy, is unclear. As *S. aureus* does not show a high antibacterial resistance in RT, other protective mechanisms, such as biofilm-formation, should be considered. Biofilm-formation is thought to be associated with antibiotic tolerance [10]. In a previous study, scanning electron microscopy showed that biofilms were present in 80% (16/20) of the recurrent tonsillitis group and in 45% (9/20) of the control group [27]. The presence of biofilms was significantly higher in the recurrent tonsillitis group, which suggests that biofilms are associated with recurrent tonsillitis [27]. The localization of the causative agents in biofilms could contribute to functional antibiotic resistance despite the absence of specific resistance mechanisms [2]. The primary problem in the treatment of patients with RT is usually difficulty in the effective eradication of the pathogen rather than its antibiotic resistance. *K. pneumoniae* is known to be a potent biofilm producer and has been isolated from tonsillar tissues [28,29].

In this study, emphasis is placed on the biological functions of bacteria in the tonsillar crypts of patients with RT and patients with a PTA. We provide a comprehensive analysis of the variety of isolated strains, *S. aureus* and *K. pneumoniae* biofilm formation, and antibacterial susceptibility with the aim of identifying any differences that may contribute to ongoing symptoms of RT or that may play a role in the development of PTAs.

## 2. Results

### 2.1. Patient Data

The age and gender ratios of the 99 patients with RT and the 29 patients with a PTA are listed in Table 1. According to our data, female patients predominated in the RT group, whereas the gender ratio was more balanced in the PTA group (Table 1). The RT group comprised 73 females (74%) and 26 males (26%), and the PTA group comprised 14 females (48%) and 15 males (52%). There was no age difference between the patients in the RT and PTA groups (independent-samples Kruskal–Wallis test, *p* = 0.617). The patients with a PTA showed increased white blood cell counts (WBCs) and C-reactive protein (CRP) levels in blood samples, and the patients with RT had WBCs and CRP levels within the normal range. These differences were highly significant.

### 2.2. Diversity of Isolated Microorganisms

Of the 128 patient samples examined, a positive cultivation finding (at least 1 pathogen or potential pathogen) was detected in 60 patients (60.6%) in the RT group and in 24 patients (82.8%) in the PTA group (Pearson χ^2^ test, *p* = 0.027). However, the greatest diversity of microorganisms was found in the patients with RT. The cultivation finding was negative; i.e., only common oropharyngeal microbiotas were cultivated in 39 patients (39.4%) in the RT group and in 5 patients (17.2%) in the PTA group (Appendix A). Irrespective of patient group, the most commonly isolated pathogenic bacterium was *S. aureus*, which was isolated as the only microorganism or co-isolated with other potentially pathogenic microorganisms (Appendix A). In the RT group, *S. aureus* was isolated in 33.3% (33/99) of cases, and, in the PTA group, it was isolated in 24.14% (7/29) of cases. Gram-positive bacteria were predominant, but at least one Gram-negative bacterium was detected in 22.2% (22/99) of patients in the RT group and in 27.6% (8/29) of patients in the PTA group. The most prevalent Gram-negative bacterium was *K. pneumoniae*; it was isolated in 10.1% (10/99) of RT cases and in 13.4% (4/29) of PTA cases (Appendix A). Moreover, *Candida* species were isolated, and they were predominant in patients with a PTA, where they were found in 48.3% (14/29) of cases and mostly as monocultures (Fisher’s test, *p* < 0.001).

### 2.3. Biofilm Growth and Associations

At least one biofilm-producing strain was found in 37.4% (37/99) of cases of RT and in 27.6% (8/29) of cases of PTA (Pearson χ^2^ test, *p* = 0.332). Moderate or strong biofilm producers were detected in 16 out of 37 cases of RT and in 2 out of 8 cases of PTA. In the RT group, among the 33 *S. aureus* isolates, 5 were strong, 8 were moderate, and 15 were weak biofilm producers, but 5 were biofilm non-producers. In the PTA group, among the 7 *S. aureus* isolates, 2 were weak biofilm producers, and 5 were biofilm non-producers. In the RT group, among the 10 *K. pneumoniae* isolates, 6 were weak biofilm producers, and 4 were biofilm non-producers. In the PTA group, among the 4 *K. pneumoniae* isolates, 1 was a weak biofilm producer, and 3 were biofilm non-producers. The biofilm mean optical densities (ODs) of all isolated *S. aureus* and *K. pneumoniae* strains are summarized in Figure 1 and Figure 2.

There was a statistically significant association found between the presence of Gram-positive bacteria and a biofilm-formation phenotype in the RT group and PTA group. If a Gram-positive microbe was present, there would most likely be a biofilm-formation phenotype (Pearson χ^2^ test, *p* < 0.001). There were no significant associations found between Gram-negative microbes, *Candida* spp., comorbidities, episodes of tonsillitis, or PTAs in medical history and a biofilm-producing strain in the RT or PTA group. There was a tendency for the PTA group to have fewer biofilm-forming strains in comparison with the RT group, although statistically significant associations were not found between the presence of biofilm-producing strains or the presence of *S. aureus* biofilm-producing strains and patient group (Table 2).

### 2.4. Antibacterial Susceptibility

*S. aureus* and *K. pneumoniae* strains are sensitive to commonly used antibiotics. One *S. aureus* isolate was identified as MRSA, which is resistant to benzylpenicillin, ampicillin, cefoxitin, ceftriaxone, ampicillin–sulbactam, and amoxicillin with clavulanic acid, but has intermediate resistance to ciprofloxacin. *S. aureus* strains resistant to benzylpenicillin, ampicillin, and at least one other antibiotic, are shown in Table 3 and Table 4 together with the biofilm-production capacity. Resistant strains are predominantly non-biofilm producers or weak biofilm producers. None of the *K. pneumoniae* isolates were extended-spectrum beta-lactamase producers.

## 3. Discussion

RT and PTAs are diseases with different clinical symptoms, disease courses, and prognoses. The common features of both diseases are as follows: they frequently occur among otolaryngology patients; their causative agent is most often a bacterium; and in cases of the ineffectiveness of antibacterial therapy, both diseases can be treated surgically. Comparably successful treatment regimens for both infections could be due to their similar etiologies [2].

*Streptococcus pyogenes* is the most common bacterial origin of acute tonsillitis in immunocompetent adults. While acute tonsillitis is postulated to only have one etiological factor, RT seems to have a multispecies etiology [5,30]. In our study, patients with RT had a great diversity of microorganisms, and polycultures were predominant. A similar finding has also been observed for other diseases. In comparison with healthy controls, increased microbial diversity is also associated with tuberculosis, cystic fibrosis, and gingivitis [12].

In our study, the isolation rate of *Streptococcus* spp. was low. Seventeen isolates of streptococci were identified in RT cases, and three were identified in PTA cases. *Streptococcus pyogenes* strains were co-isolated in four patients with RT, and none were isolated in patients with a PTA. Other studies have also reported a low isolation rate (1.7–5%) of streptococci in patients with RT; streptococci have been found to be less prevalent in the tonsillar core (1.7%) [29,31] than on the tonsillar surface. *S. pyogenes* in the RT pathogenesis has most likely been overrated or, alternatively, decreased in recent years [2,32].

The isolation rate of microorganisms varies depending on the approaches used for material collection. In a report by Zautner et al., *S. aureus* was prevalent in patients with RT (57.7%), but *S. pyogenes* was prevalent in patients with a PTA (in 20.2% of tonsillar cell suspensions) [2]. In a study conducted by Vaikjarv et al., it was demonstrated that tonsillar fossa biopsy specimens were better materials for microbiological analyses than abscess pus samples, because they revealed more bacteria per culture [14]. *Streptococcus* spp. were the most common bacteria found in tonsillar fossa biopsy specimens and pus samples, but *Staphylococcus* spp. were also found in tonsillar fossa biopsy specimens, and staphylococci were not found in any pus cultures [14]. We chose to analyze biopsy samples to make the RT and PTA patient groups more comparable and inoculations more informative.

In our study, four *Streptococcus anginosus* isolates were found in the RT group, and none were found in the PTA group. In contrast, in another study, bacteria from the *Streptococcus anginosus* group were detected in the patient samples of the PTA renewal group more often than in those of the PTA recovery group, and the authors concluded that bacteria in the *Streptococcus anginosus* group appear to predict the renewal of PTA symptoms [33].

On the contrary, our study showed a high rate of *Candida* spp. They were isolated in 8.08% (*n* = 8) of patients with RT and in 48.23% (*n* = 14) of patients with a PTA. Several publications have claimed a lower isolation rate of *Candida* spp. In a report by Katkowska et al., *Candida* spp. were found in tonsillar core samples from patients with RT at a rate of 2.5%, in tonsillar surface samples at a rate of 8.3%, and in throat samples at a rate of 9.3% before tonsillectomy [5]. Zautner et al. reported *Candida* spp. in the tonsillar tissues of patients with RT at a rate of 12.8%, but, in patients with a PTA, this rate was 4.9% [2]. Slouka et al. reported yeasts in 2.3% of PTA pus aspirates [7]. Our study results are in agreement with those of a study conducted by Jokinen et al. [34], wherein fungal cultures of the tonsils of 147 patients with chronic tonsillitis revealed *Candida albicans* in 41.4% of cases; in the control group of healthy individuals, the rate of *Candida albicans* was 51–5% (34). The pathogenicity of the fungi was investigated in each case by histological means. The histological investigations revealed no evidence of pathogenicity in these organisms because they were found in the tonsillar crypts, and no granulomatous inflammation was seen surrounding them [34]. Another publication compared the palatine tonsil mycobiomes between individuals with human immunodeficiency virus (HIV) and those without it [35]. It was found that, between the individuals with HIV and those without it, in contrast to the bacteriomes, the palatine tonsil mycobiomes did not differ significantly between the two groups [35]. The role of *Candida* spp. in the tonsillar inflammatory process cannot be convincingly judged based on the results of our study, and it should be clarified in future studies.

In the scope of the present study, Gram-negative microorganisms were not the most important. *K. pneumoniae* is known to be a potent biofilm producer [28]. In our study, not only did *K. pneumoniae* have a low incidence, but it also showed low biofilm-producing ability. None of the *K. pneumoniae* isolates were a strong biofilm producer. *H. influenzae* is known to be more frequently isolated from the tonsillar core. In our study, it was isolated from two patients with RT, indicating that it is much less prominent than previously described. There was no statistically significant association between the presence of Gram-negative microbes and the presence of biofilm-producing strains in tonsillar tissues, in patients with RT, or in patients with a PTA.

In our study, the median age at the time of PTA occurrence was 31 years, which is in accordance with the results of other studies where patients were 29–34 years old [36,37]. In a previous study, an older population (over 40 years of age) with PTA was found to present with significantly lower rates of aerobic bacteria and a tendency toward higher rates of anaerobic growth; the authors clarified the need for prompt and aggressive surgical and antibiotic treatment for older patients with a PTA [38]. Group A *streptococcus* spp. was significantly more frequently recovered from patients with a PTA in the winter and spring than in the summer [22]. No seasonal differences were observed in our study. Blood of patients with PTA had higher WBCs and CRP levels compared to the blood samples of patients with RT. Patients with PTA were included in the study during the acute inflammation stage, which explains the high inflammatory parameters. Patients with RT had WBCs and CRP levels within normal range, but WBCs were at the lower limit of normal range. RT patients may have changes in peripheral blood samples. Other studies had analyzed cytokine production, T and B lymphocytes, and the neutrophil-lymphocyte ratio in blood samples of patients with RT to draw reliable conclusions [39,40].

*S. aureus* isolates showed resistance to benzylpenicillin and ampicillin, which is consistent with the results of other studies. In a study conducted by Katkowska et al., *S. aureus* isolates from patients with RT showed resistance to penicillin in 79% of cases and to ampicillin in 63.2% of cases; only one *S. aureus* isolate was MRSA [5]. In our study, we did not observe associations between antibiotic resistance and biofilm-formation intensity. Even though several studies have claimed that intensive biofilm formation is associated with increased antibiotic tolerance, Ma et al. demonstrated that increased biofilm formation had the opposite effect and resulted in less antibiotic tolerance [41]. Our study data do not support the hypothesis that patients with a PTA are more likely to have strong biofilm-producing strains than other strains. On the contrary, in the PTA group, among the seven *S. aureus* isolates, two were weak biofilm producers, and five were biofilm non-producers. These findings suggest that *S. aureus* isolates could be wild-type strains and not an endogenous infection in the PTA group. Patients with RT who have PTAs in their medical history are more likely to have a biofilm-producing phenotype (*p* = 0.091). It is likely that, if a larger number of patients were included in this study, then the association would have been confirmed.

A biofilm is described as a bacterial community wrapped in a self-produced matrix of extracellular polymeric substances [42]. Studies have been carried out on the interactions between microorganisms. Interactions between *C. albicans* and *Staphylococcus* spp. are apparently synergistic or mutualistic, and they are increasingly reported [43]. The findings suggest that fungal cells can modulate the action of antibiotics and that bacteria can affect antifungal activity in mixed fungal–bacterial biofilms [44]. Prostaglandin E2 from *Candida albicans* stimulates the growth of *S. aureus* in mixed biofilms [45]. A chronic inflammatory process in palatine tonsils could be explained by the protected status of a bacterial community and a great variety of microorganisms.

## 4. Materials and Methods

A total of 128 patients were enrolled in this prospective, monocentric study. A total of 29 patients were diagnosed with PTAs, and 99 patients had RT. Study patients underwent tonsillectomy in Pauls Stradins Clinical University Hospital in the period of 2018–2020.

The inclusion criteria were as follows: a diagnosis of PTA or RT, bilateral tonsillectomy received as surgical treatment, and no antibacterial therapy for at least 4 weeks. RT was defined as at least seven attack episodes per year in the preceding one year, five episodes per year in the preceding two years, or three episodes per year in the preceding three years despite adequate antibiotic therapy. PTA was defined clinically by redness and arching of the palpably stiff and markedly painful soft palate and fever, a sore throat, unclear speech, sometimes trismus, and a reaction of the descending lymph nodes. All RT patients underwent scheduled tonsillectomy, with the last episode of tonsillitis being more than 4 weeks ago. All PTA patients underwent immediate tonsillectomy under acute infection stage.

The exclusion criteria were as follows: hematologic system diseases (thrombocytopenia or coagulopathy), primary or acquired immunodeficiency, antibacterial therapy received in the last 4 weeks or the initiation of antibacterial treatment prior to material collection, outpatient treatment, other methods of PTA treatment such as incision and drainage, and failure to perform tonsillectomy (contraindications for surgery or general anesthesia) or to obtain written informed consent. Patients with active solid or hematological malignancy, active autoimmune disease, usage of immunosuppressive agents prior to admission (prednisone > 10 mg/day or equivalent) more than 4 weeks, and solid organ transplant, were excluded.

Pediatric patients were not excluded from the study since only adult patients (18 years old or older) are treated at Pauls Stradins Clinical University Hospital. Pediatric patients receive medical treatment at the Children’s Clinical University Hospital in Riga, Latvia.

After tonsillectomy, a histological analysis of the specimens from all patients was routinely performed to verify the clinical diagnosis.

The study protocol was approved by the local Ethics Committee of Riga Stradins University (document No. 49/30.11.2017.), and all of the data were collected according to the guidelines on data protection and confidentiality. Written informed consent was obtained from all subjects before the study.

### 4.1. Sample Collection

Samples for microbiological testing were obtained from tonsillar crypts with a punch-biopsy needle perioperatively [17,46]. The samples contained specimens of affected tonsillar tissue in the case of a PTA.

### 4.2. Isolation of Microorganisms and Microbiological Investigation

Punch-biopsy samples of tonsillar crypts were taken, placed in Amies transport media, and transported at room temperature within 24 h to the laboratory. The samples were cultivated on Columbia blood agar, Mannitol salt agar, MacConcey agar, and Sabouraud dextrose agar plates at 36 ± 1 °C for 24–48 h aerobically. A brucella blood agar plate in an anaerobic pouch system, incubated at 36 ± 1 °C for up to five days, was used for the cultivation of anaerobes. A Columbia can blood agar plate with an optochin disc incubated in a CO_2_ incubator at 36 ± 1 °C for 24–48 h was used for the cultivation of Streptococcus pneumoniae, and a chocolate agar plate with an oleandomycin disc incubated in an CO_2_ incubator at 36 ± 1 °C for 24–48 h was used for the cultivation of *Haemophylus* spp.

We took note of the common oropharyngeal microbiota as described by the European Society of Clinical Microbiology and Infectious Diseases [47]. The identification of the considered pathogens was performed using a Microflex LT (Bruker Daltonics flexAnalysis version 3.4, Bruker Daltonics GmbH & Co. KG, Bremen, Germany) matrix-assisted laser desorption ionization–time-of-flight mass spectrometer (MALDI–TOF MS) system.

Susceptibility testing and the evaluation of the results were performed using the disc diffusion method, and the evaluation of the results was carried out according to European Committee on Antimicrobial Susceptibility Testing (EUCAST) standard actual EUCAST version [48].

### 4.3. Biofilm Growth Using Cristal Violet Assay

Isolated Gram-positive strains were suspended in trypticase soy broth (TSB) supplemented with additional 1% glucose, and Gram-negative strains were suspended in Luria–Bertani (LB) broth for incubation at 37 °C for 16–18 h. Inoculated broths were diluted with sterile TSB or LB broths in a ratio of 1:100. Then, 150 µL of the diluted suspension was transferred with a multichannel pipette in sterile 96-well plates (Thermo Scientific™ Nunc MicroWell 96-Well Microplates, flat bottom, Thermo Fisher Scientific, Roskilde, Denmark). Each plate contained 11 strains, and the negative control (uninoculated broth) contained 8 wells per strain; each experiment was performed in triplicate. The inoculated plates were cultivated aerobically at 37 °C for 48 h. After incubation, all wells were emptied by gently throwing out the liquid in a clinical waste bag without the use of a pipette. Each well was rinsed 3 times with sterile 250 µL 0.9% saline. After washing, staining was performed by adding 150 µL of 0.1% crystal violet per well. After 15 min, the color was removed by gently throwing out the color, and each well was washed 3 times with 250 µL distilled water. At the end, 150 µL of 96% ethanol was added to each well. Afterwards, the optical densities (ODs) of the wells were measured at a 570 nm wavelength with a microplate spectrophotometer (Tecan Infinite F50, Mannedorf, Switzerland, with Magellan™ reader control and data analysis software V 6.6) [49].

### 4.4. Biofilm Calculation

The OD values for each strain were averaged and are expressed as numbers. The cut-off value (ODc) was defined as three standard deviations (SDs) above the mean OD of the negative control, and it was separately calculated for each plate. Strains were divided as follows: OD ≤ ODc = no biofilm producer, ODc < OD ≤ 2 × ODc = weak biofilm producer, 2 × ODc < OD ≤ 4 × ODc = moderate biofilm producer, and 4 × ODc < OD = strong biofilm producer [50].

### 4.5. Statistical Analysis

A statistical analysis was performed using IBM SPSS Statistics version 26 (Chicago, IL, USA) and Microsoft Excel 10. For all of the hypotheses tested, a *p*-value of less than 0.05 indicated statistical significance.

## 5. Conclusions

### 5.1. Conclusions

*S. aureus* is the most common pathogen isolated from patients with RT, and *Candida* spp. are the most common pathogens isolated from patients with a PTA.

*S. aureus* isolates associated with RT and PTAs are susceptible to most antibiotics.

Patients with RT more commonly have biofilm-producing strains, but patients with a PTA have biofilm-non-producer strains as causative agents.

*K. pneumoniae* does not play a major role in biofilm production.

### 5.2. Strengths and Limitations

Several limitations should be addressed. Firstly, we analyzed microorganisms separately, and it would be desirable to analyze their interactions, as well as the functional aspects of the biofilm. Secondly, further studies are necessary to evaluate the role of *Candida* spp. *S. aureus* surface proteins and to genotype the pathogenesis of tonsillitis.

## 6. Patents

Klagisa, R.; Kroica, J.; Kise, L. Punch Biopsy Needle. Patent No: LVP2020000055. In *Izgudrojumi, Preču Zīmes un Dizain-paraugi*. Patent Office of the Republic of Latvia, Riga, Latvia, 2021; Volume 5, p. 315.

## Figures and Tables

**Figure 1 ijms-23-10273-f001:**
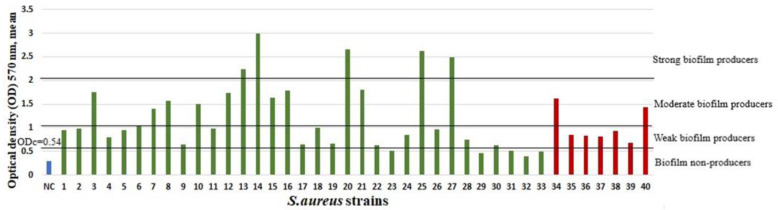
Biofilm-production capability on a microtiter plate of *S. aureus* strains from patients with RT (1–33 green bars) and patients with a PTA (34–40 red bars). Bars represent mean values of OD (measured at wavelength of 570 nm). Trypticase soy broth with 1% glucose was used as a negative control (NC, blue bar). The cut-off value (ODc) and biofilm-production-capacity levels are marked with horizontal lines.

**Figure 2 ijms-23-10273-f002:**
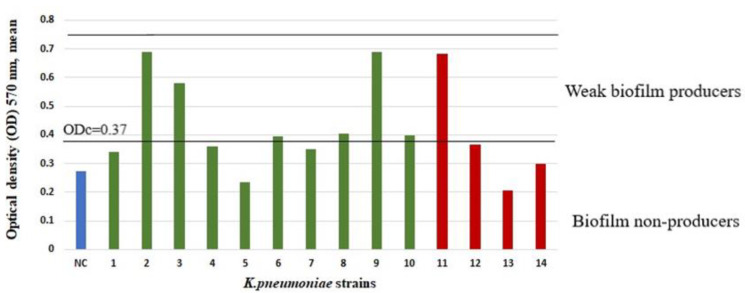
Biofilm-production capability on a microtiter plate of *K. pneumoniae* strains from patients with RT (1–10 green bars) and patients with a PTA (11–14 red bars). Bars represent mean values of OD (measured at wavelength of 570 nm). Luria–Bertani medium was used as a negative control (NC, blue bar). The cut-off value (ODc) and biofilm-production-capacity levels are marked with horizontal lines.

**Table 1 ijms-23-10273-t001:** General characteristics of the study population.

Characteristics		RT	PTA	*p*-Value
Gender	Male, *n* (%)	26 (26%)	15 (52%)	*p* = 0.061
	Female, *n* (%)	73 (74%)	14 (48%)	*p* = 0.061
Age	Age (between, mean ± SD), years	20–72, 32.94 ± 11.19	18–58, 32.4 ± 12.2	*p* = 0.279
	Age (median, IQR), years	31, 10	31, 16	
Laboratory findings	CRP, median, mg/L	1.17	85.5	*p* < 0.001
	WBC, median, ×10^9^/L	6.52	12.97	*p* < 0.001
Comorbidities	Primary arterial hypertension (*n*)	9	1	*p* = 0.454
	Cardiologic diseases (*n*)	5	0	*p* = 0.587
	Type 2 diabetes mellitus (*n*)	1	0	*p* > 0.999
	Bronchial asthma (*n*)	5	1	*p* > 0.999
	Chronic gastritis or gastroesophageal reflux disease (*n*)	17	1	*p* = 0.188

**Table 2 ijms-23-10273-t002:** Comparison of patients’ microbiological data in the RT and PTA groups.

Patients’ Microbiological Data		RT Group	PTA Group	*p*-Values
Isolation rate	*S. aureus, n* (%)	33/99 (33.33%)	7/29 (24.14%)	*p* = 0.347
	*K. pneumoniae, n* (%)	10/99 (10.10%)	4/29 (13.79%)	*p* = 0.519
	*Candida* spp., *n* (%)	8/99 (8.08%)	14/29 (48.28%)	*p* < 0.001
Biofilms, mean OD	*S. aureus* biofilms, mean OD	1.24	1.02	*p* = 0.929
	*K. pneumoniae* biofilms, mean OD	0.44	0.39	*p* = 0.322
Biofilm-producing strains	Biofilm-producing strains, *n*	37	8	*p* = 0.111
	*S. aureus* biofilm-producing strains, *n*	28	7	*p* = 0.642
	*S. aureus* moderate and strong biofilm producers, *n* (%)	13/33 (39.39)	2/7 (28.57%)	*p* = 0.691
	*K. pneumoniae* moderate and strong biofilm producers, *n*	0	0	
Associations between variables by study groups	Gram-positive microbe and biofilm-producing strain	*p* < 0.001	*p* < 0.001	
	Gram-negative microbe and biofilm-producing strain	*p* = 0.227	*p* > 0.999	
	*Candida* spp. and biofilm-producing strain	*p* > 0.999	*p* = 0.215	
	Comorbidities and biofilm-producing strain	*p* = 0.759	*p* = 0.540	
	Episodes of tonsillitis and biofilm-producing strain	*p* = 0.313	*p* = 0.738	
	PTA in medical history and biofilm-producing strain	*p* = 0.091	*p* = 0.640	

**Table 3 ijms-23-10273-t003:** Antibiotic-susceptibility and biofilm-production-ability patterns of *S. aureus* strains.

Antibiotics	*S. aureus* Strains (*n* = 33) of Patients with RT			*S. aureus* Strains (*n* = 7) of Patients with a PTA		
	Resistant Strains (*n*)	Non- and Weak Biofilm Producers (*n*)	Moderate and Strong Biofilm Producers (*n*)	Resistant Strains (*n*)	Non- and Weak Biofilm Producers	Moderate and Strong Biofilm Producers
P, AMP, CIP *	20/33	12/20	8/20	5/7	4/5	1/5
P, AMP, CIP *, CD *	1/33	1				
CIP *	9/33	5/9	4/9	2/7	1/2	1/2
P, AMP, CIP *, E	1/33		1			
CIP *, E	1/33	1				
FOX, CRO, P, AMP, AMS, AUG, CIP *	1 **/33	1				

Note: *, intermediate resistance; **, MRSA; FOX, cefoxitin; CRO, ceftriaxone; P, benzylpenicillin; AMP, ampicillin; AMS, ampicillin–sulbactam; AUG, amoxicillin–clavulanic acid; CIP, ciprofloxacin; E, erythromycin; CD, clindamycin. Each antibiotic resistance was determined separately.

**Table 4 ijms-23-10273-t004:** Antibiotic-susceptibility and biofilm-production-ability patterns of *S. aureus* strains by patient group.

Patient Group	Biofilm Formation	Antibiotic Resistance (P, AMP)	No Antibiotic Resistance (or Antibiotic Resistance to One Antibiotic)	*p*-Value
RT group	Non- or weak biofilm producer	14	6	*p* = 0.590
	Moderate/strong biofilm producer	9	4	*p* > 0.999
PTA group	Non- or weak biofilm producer	4	1	*p* > 0.999
	Moderate/strong biofilm producer	1	1	*p* > 0.999
PTA + RT group	Non- or weak biofilm producer	18	7	*p* = 0.590
	Moderate/strong biofilm producer	10	5	*p* > 0.999

Note: P, benzylpenicillin; AMP, ampicillin. Each antibiotic resistance was determined separately.

## Data Availability

The datasets generated are available from the corresponding author upon reasonable request.

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
