# Peer review of "Analysis of Microorganism Colonization, Biofilm Production, and Antibacterial Susceptibility in Recurrent Tonsillitis and Peritonsillar Abscess Patients"

_ijms, 2022, doi:10.3390/ijms231810273_

Round 1

Reviewer 1 Report

1.      In page 10, line 303, the definition of PTA and RT you included in your study is needed.

2.      In page 10, line 304, you mentioned "surgical treatment". Do you mean quinsy tonsillectomy for PTA/RT under acute infection stage, or do you mean tonsillectomy after acute infection stage for definite treatment to prevent recurrence? If the surgery is not arranged in acute stage, then what’s the time interval between last infection? If both conditions are enrolled, then please tell us about the ratio in each group.

3.      In page 10, line 311: typing error, double “(“.

4.      In page 3, section 2.1 patient date, please give more detailed information about patient’s underline disease, such as diabetes.

5.      In page 3, table1, the WBC in RT group is within normal limits, and quite low. Please give some discussion about this situation.

6.      In page 3, table1, since you provided the median value of patient’s age, maybe it will be nice if you can provide IQR as well.

7.      In table 1, all patients are older than 20 in RT group and older than 18 in PTA group. Did you exclude pediatric patients? If yes, you will need to demonstrate it in your methodology section. We know this will influence the pathogenic bacteria in tonsillitis patients.

8.      Please consider if you want to move table 2 and 3 to supplemental data.

9.      Table 4 is an important table for your manuscript. Please make the table more structural and easier to understand. I don’t think the statistical method you use is interesting enough to be demonstrated in this table.

10.  Your title is “Novel Technique for Sample Collection to Investigate Causative Agents of Recurrent Tonsillitis and Peritonsillar Abscesses”, however you barely discussed the “novel technique” in your discussion section. Usually when the title is emphasizing “novel technique”, we will want to compare results between novel technique and traditional technique. Either you will want to improve the study design and discussion, or simply adjust your title.

Reviewer 2 Report

Novel Technique for Sample Collection to Investigate Causative Agents of Recurrent Tonsillitis and Peritonsillar Abscesses

By Klagisa et al.

The paper analyzes the causative agents for Recurrent Tonsillitis (RT) and Peritonsillar Abscesses (PTA), their antibiotic resistance and correlation, or lack thereof with biofilm formation. The paper presents interesting observations on a very important topic. The paper is mostly well-written, except for lack of clarity in some places as pointed out below.

Page 2 Line 92: “role of the former pathogen” Not clear what is meant by “former”. There is only one pathogen mentioned, S. aureus.

Page 2 Line 93: “As S. aureus does not show a high antibacterial resistance in RT, other mechanisms of resistance, such as biofilm-formation, should be considered.” Not clear what this means. Is there resistance or is there not? Both cannot be true. If there is little or no resistance, why does the question of mechanism of resistance arise?

Page 3 Line 107: “the formation of S. aureus and K. pneumoniae biofilms, S. aureus biofilm formation” Not clear why the same statement is written twice in the sentence.

Page 9 line 274: “The authors explained this as a result of the bacteria having a higher rate of metabolic activity compared to the normal decreased metabolic state of biofilms [39].” It is good that the results in this manuscript agree with the results in the cited paper (which I have not read). However, the reason stated, is the opposite of conventional wisdom. I thought, low metabolic activity make bacteria more resistant to antibiotics because most commercial antibiotics preferentially kill growing bacteria although there have been recent reports of some antibiotics preferentially killing non-growing or slow-growing bacteria.

Page 11 Line 364: “it was separately calculated for each experiment.” Not clear what is meant by each experiment. There are two figures, so were there two experiments?

Table 2: Last row: CIP is broken into two lines (C and IP). Please keep them in one line.

Table 5: In the first column (Antibiotics), the meaning of the + sign is not clear. For example, for P+AMP+CIP, were the three antibiotics added together or independently? I assume, they were not present together, in which case, I think, the + should be replaced with a comma (,) or it can be mentioned in the footnote that each antibiotic resistance was determined separately. Same comment for Table 6.

Table 5 column 1. Since MRSA is not the name of an antibiotic, it should not be written in this column.  I think the best place for this is as a footnote for the number “1” in the next column.

Table 5 and also in line 179: “MRSA” Methicillin is not listed in the Antibiotics column. So how is this strain classified as MRSA? If methicillin was tested, why is it not in the list in the first column? Also in Line 179, “One S. aureus isolate was identified as MRSA” How was it identified? Was it identified by PCR for mecA or any other gene? Is the term “MRSA” being used synonymously with multi-drug resistant bacteria?

Table 6: Why is the column 2 heading “Antibiotic susceptibility”? This column is about biofilm.

Table 5 and 6 Titles are almost exactly the same except that Table 6 mentions “by patient group”. The data in Table 5 also arranges by patient group. Are these two tables, based on the same dataset and can we find the numbers of Table 6 in Table 5?

Table 6 Column 3: Antibiotic resistance is mentioned as P + AMP, but this combination is not found in Table 5, not clear why.

Round 2

Reviewer 1 Report

The manuscript makes much more sense after the authors adjusted their title. I'm still concerned about the fact that they are comparing chronic/subacute phase RT and acute phase PTA. Methodology is questionable here, but there is obviously no way for them to fix it. I hope the authors can find some reasonable references to support their study design.

I appreciate the authors hard work for replying our comments thoroughly. There are some points I am still concerned about that might improve the work.

1. In page 13, line 318, you enrolled all patients of peritonsillar abscess who received surgery. In line 325, you mentioned that all PTA patients received immediate tonsillectomy under acute phase. This is interesting because the most widely accepted first line treatment for PTA should be incision and drainage, which is usually very effective for the majority of PTA patients. Please confirm you statement means that incision and drainage was never arranged in your hospital during this period of time.

2. Your revised version pointed out you are actually comparing chronic/subacute phase RT and acute phase PTA (Page 13, line 324~326). This is a very questionable methodology.  Please give some references to support your work.

3. In point 7 of first round of revision, you mentioned "We did not exclude pediatric patients since only adult patients (≥ 18 y.o.) are treated in P.Stradiņš Clinical University Hospital. Pediatric patients receive medical treatment in Children`s Clinical University Hospital." I believe you should add this description to your manuscript.

Reviewer 2 Report

The authors have addressed all my concerns. The manuscript is now much better.

Round 3

Reviewer 1 Report

I believe all the comments were responded thoroughly. The work is much improved.